# Functions of PPR Proteins in Plant Growth and Development

**DOI:** 10.3390/ijms222011274

**Published:** 2021-10-19

**Authors:** Xiulan Li, Mengdi Sun, Shijuan Liu, Qian Teng, Shihui Li, Yueshui Jiang

**Affiliations:** School of Life Sciences, Qufu Normal University, Qufu 273165, China; sunzhuzhu12138@163.com (M.S.); sjliusj@163.com (S.L.); T19819061626@163.com (Q.T.); lshih4030@163.com (S.L.)

**Keywords:** PPR protein, cytoplasmic male sterility, seed development, RNA editing, RNA splicing

## Abstract

Pentatricopeptide repeat (PPR) proteins form a large protein family in land plants, with hundreds of different members in angiosperms. In the last decade, a number of studies have shown that PPR proteins are sequence-specific RNA-binding proteins involved in multiple aspects of plant organellar RNA processing, and perform numerous functions in plants throughout their life cycle. Recently, computational and structural studies have provided new insights into the working mechanisms of PPR proteins in RNA recognition and cytidine deamination. In this review, we summarized the research progress on the functions of PPR proteins in plant growth and development, with a particular focus on their effects on cytoplasmic male sterility, stress responses, and seed development. We also documented the molecular mechanisms of PPR proteins in mediating RNA processing in plant mitochondria and chloroplasts.

## 1. Introduction

Pentatricopeptide repeat (PPR) proteins are characterized by the presence of tandem arrays of a degenerate 35-amino-acid repeat motif, PPR motif [1]. Based on the types of motif and their arrangement, PPR proteins are divided into two classes, P and PLS. P-class proteins only contain canonical P-motifs with 35 amino acids, whereas PLS-class proteins consist of P-, L- (35 or 36 amino acids), and S- (31 or 32 amino acids) motifs forming tandemly repeated PLS triplets [2]. Many of the PLS-class proteins are carboxyl terminally extended with highly conserved E, E+, or DYW domains. Thus, PLS-class proteins can be further divided into PLS, E, E+, and DYW subclasses according to the domains identified in carboxyl terminal [3].

PPR proteins are sequence-specific RNA-binding proteins that are mostly localized to mitochondria and/or chloroplasts, where they are involved in RNA post-transcriptional processing [4] (Figure 1A). In general, the P-class PPR proteins mediate diverse aspects of RNA processing in plant organelles, while the PLS-class PPR proteins mainly function in RNA editing [5]. Mutations in these PPR protein-coding genes lead to the dysfunction of mitochondria and/or chloroplasts, thereby resulting in growth retardation, pollen abortion, and seed development defects in plants [4], indicating the important roles of PPR proteins in plant growth and development (Figure 1B).

## 2. Functions of PPR Proteins in Cytoplasmic Male Sterility

Cytoplasmic male sterility (CMS) is a maternally inherited trait that presents a defect in the production of viable pollen. CMS is widespread in higher plants, and has been widely used in the production of hybrid seeds and utilization of heterosis in many crop species [6,7]. Plant CMS is usually caused by mutations, rearrangements, or recombinations of mitochondrial DNA, and in many instances, male fertility can be restored specifically by restorer-of-fertility (*Rf*) genes in the nuclear genome [8,9]. To date, more than ten *Rf* genes have been cloned and functionally characterized in various crop species, and the majority of them were found to encode PPR protein, including *Rf1a* [10], *Rf1b* [11], *Rf3* [12], *Rf4* [13], *Rf5* [14], *Rf6* [15] in rice; *Rfo* [16], *PPR-B* [17], *RsRf3-4* [18], *Rfk1* [19] in radish; *Rf1* [20], *Rf2* [21] in sorghum; *Rfp* [22], *Rfn* [23] in rapeseed; *Rf-PPR592* [24] in petunia; *BrRfp1* [25] in Chinese cabbage; and *Rfm1* [26] in barley. With the exception of the sorghum *Rf1* [20] and the barely *Rfm1* [26], the PPR-type *Rf* genes encode PPR proteins belonging to P class.

Members of P-class PPR proteins mostly function in various post-transcriptional process of organellar RNAs, such as RNA splicing, RNA stabilization, RNA cleavage, and translation [4,5]. Proteins encoded by *Rf* genes usually target mitochondria and act as fertility restorers by suppressing the production of mitochondrial CMS-inducing proteins [6,11]; however, the exact molecular mechanisms underlying fertility restoration by RF proteins is presently unclear. The PPR-type RF proteins have been proposed to rescue fertility by regulating the expression of CMS-conferring genes through the similar molecular mechanisms as that of other PPR proteins. In most of the CMS systems, PPR-type RF proteins bind to the mitochondrial CMS-conferring transcripts and promote their cleavage or degradation [27]. For example, rice RF1A and RF1B proteins, respectively encoded by *Rf1a* and *Rf1b* genes, have been considered to restore CMS by processing mitochondrial *orf79* transcript via different mechanisms. RF1A directly binds to and cleaves the *atp6-orf79* transcript at the intercistronic region, whereas RF1B promotes the rapid degradation of the *atp6-orf79* transcript [10,11]. The protein encoded by rice *Rf4* gene was reported to suppress *WA352*-mediated male sterility by reducing *WA352* transcript levels [13]. Two rice RF proteins, Rfp and Rfn, are involved in transcript cleavage of *orf224* and *orf222*, respectively [22,23].

In some CMS systems, PPR-type RF proteins restore the male fertility by impeding the translation or post-translational processing of mitochondrial CMS-inducing proteins. Studies on radish PPR-B revealed that PPR-B protein does not function through cleavage or degradation of the *orf138* mRNA, but rather block its translation by inhibiting either the association with or the progression of mitochondrial ribosomes on the *orf138* mRNA [17]. Similarly, rice RF3 protein does not affect the abundance of *WA352* transcript but suppresses the accumulation of WA352 protein [12]. In addition, *Rf1* from sorghum and *Rfm1* from barely encode PPR proteins belonging to PLS class. As PLS-class PPR proteins almost exclusively play a role in RNA editing, it is possible that sorghum *Rf1* and barely *Rfm1* restore pollen fertility by editing *S-orf* transcripts or other target RNAs [8,26].

CMS was not only an ideal model system to study the interaction between mitochondrial and nuclear genomes but also a useful genetic tool for breeding to exploit hybrid vigor in crops [27]. Although PPR proteins are involved in the restoration of male fertility, functions of most PPR proteins are still obscure. Therefore, exploring the functions of PPR proteins will contribute to understanding the CMS mechanism and improving molecular breeding in crops.

## 3. Functions of PPR Proteins in Plant Responses to Biotic and Abiotic Stresses

In previous studies, PPR genes were found to change their expression patterns under biotic and abiotic stresses and regulate growth in many plants. For instance, in Arabidopsis, 11 PPR proteins have been shown to respond to biotic or abiotic stresses. SOAR1, a cytosol-nucleus dual-localized PPR protein, is involved in ABA signaling and tolerance to drought, salt, and cold stress [28]. GUN1 is a chloroplast-located PPR protein, the *gun1* mutant is defective in response to norflurazon, lincomycin, and high-light treatments [29], and it also exhibits a more susceptible phenotype to photooxidative stress caused during the de-etiolation [30]. In addition, nine mitochondria-located PPR proteins, MED11/LOI1 [31], PPR40 [32], ABO5 [33], PGN [34], AHG11 [35], SLG1 [36], SLO2 [37], PPR96 [38], and POCO1 [39] were reported to participate in responses to many abiotic or biotic stresses. In rice, two chloroplast-located PPR proteins, OsV4 and TCD10, are required for chloroplast development at early seedling stage under cold stress [40,41]. The chloroplast-located PPR protein WSL affects chloroplast development and abiotic stress response in rice, and the *wsl* mutant displays chlorotic striations early in development and enhanced sensitivity to ABA, salt, and sugar [42]. Recently, a mitochondrial PPR protein OsNBL3 was found to be involved in response to biotic or abiotic stresses. The *nbl3* mutant exhibits growth retardation, leaf wilting, and premature senescence, and it shows enhanced resistance against fungal and bacterial pathogens and to salt stress [43].

A few of the PPR proteins involved in plant responses to biotic and abiotic stresses have been shown to play roles in post-transcriptional processing of RNA in mitochondria and chloroplasts. In Arabidopsis, PGN participates in editing of mitochondrial *cox2* and *nad6* [34]. SLG1 and AHG11 are involved in editing of mitochondrial *nad3* and *nad4*, respectively [35,36]. SLO2 was found to be required for mitochondrial RNA editing at multiple sites [37]. In rice, WSL functions in the splicing of chloroplast *rpl2* [42]. OsNBL3 takes part in the splicing of mitochondrial *nad5* [43]. Although *POCO1* encodes a mitochondria-localized P-class PPR protein, multiple RNA editing defects were identified in *poco1* mutant, which suggest that POCO1 may be required for RNA editing [39].

Under normal and stress conditions, retrograde signaling from organelles plays vital roles in coordinating the expression of nuclear and organellar genes, and in regulating plant growth and development. Arabidopsis GUN1 is identified to be a central integrator of chloroplast retrograde signaling pathways. In 2016, Tadini et al. revealed that GUN1 controls accumulation of the chloroplast ribosomal protein S1 at the protein level and interacts with proteins involved in chloroplast protein homeostasis [44]. MORF2 is an essential component of RNA editosome and is required for editing at almost all sites in chloroplast RNAs [45]. Unlike typical PPR proteins, GUN1 does not appear to bind to RNAs. However recently, GUN1 was found to physically interact with MORF2 to regulate the editing efficiency of multiple sites within chloroplast RNAs and modulate the activity of the nucleus-encoded chloroplast RNA polymerase, particularly during retrograde signaling [46,47], indicating that GUN1 is important for chloroplast RNA metabolism and chloroplast-to-nucleus retrograde communication. The mitochondrial PPR protein LOI1 of Arabidopsis was reported to be involved in RNA editing of mitochondrial transcripts *cox3*, *nad4*, and *ccb203*, and regulate biosynthesis of isoprenoids, metabolites known to affect defense gene expression in response to wounding and pathogen infection [31]. The *loi1* mutant has decreased sensitivity to two inhibitors of isoprenoid synthesis, fungal phytotoxin lovastatin, and herbicide clomazone, showing the indirect effects of retrograde signaling from mitochondria to the cytoplasm to evoke alteration of the mevalonate pathway [48]. Therefore, these results reveal a putative link between organellar RNA processing and plant responses to environmental stresses.

## 4. Functions of PPR Proteins in Seed Development

In angiosperm plants, seed development starts with double fertilization of egg and central cells with two sperm cells, which leads to the formation of a diploid embryo and a triploid endosperm, and develops into mature seeds comprising three structures: maternal coat, embryo, and endosperm [49]. Development of embryo and endosperm is well correlated and regulated by numerous distinct proteins involved in many important physiological processes [50], including cell growth, RNA transcription and post-transcriptional processing, etc.

In recent years, more and more genetic and biochemical studies have shown that PPR proteins play important roles in seed development of higher plants, and loss-of-function of these PPR proteins usually leads to defects in embryogenesis and/or endosperm development [4,51]. According to the phenotypic expression, seed mutants can be divided into four major classes: *empty pericarp* (*emp*), *embryo specific* (*emb*), *defective kernel* (*dek*), and *small kernel* (*smk*). A detailed summary of the maize and Arabidopsis seed mutants caused by the functional defects of PPR proteins is provided in Table 1.

Most PPR proteins identified to date are targeted to mitochondria and/or chloroplasts [4]. The disruption of PPR proteins localized to chloroplasts usually results in the *emb* mutants that are defective in embryogenesis, but relatively normal in endosperm development. For instance, PPR8522 [94] and EMB-7L [95] in maize and GRP23 [103] in Arabidopsis are necessary for embryogenesis, and their mutations lead to arrested embryo development at the transition stage, resulting in an embryo-lethal phenotype. For mitochondrion-targeted PPR proteins, their disruptions mostly cause diverse seed development mutants, including *smk*, *dek*, and *emp*, with different degrees of defects in embryo and endosperm. The loss-of-function of SMK1 [65], SMK4 [66], SMK6 [67], ZmSMK9 [68], PPR2263 [69], and MPPR6 [70] in maize and PPR19 [97] in Arabidopsis arrests both embryo and endosperm development, resulting in *smk* phenotypes. Some characterized mitochondrion-targeted PPR proteins, such as DEK2 [52], DEK10 [53], DEK35 [54], DEK36 [55], DEK37 [56], DEK39 [57], DEK40 [58], DEK41/43 [59,60], DEK46 [61], and DEK53 [62] in maize, are necessary for seed development, and their disruptions result in *dek* mutants with arrested development of both the embryo and endosperm at an early stage. Meanwhile, many PPR proteins are targeted to mitochondria and function in development of both embryo and endosperm, and mutations in their encoding genes arrest embryo and endosperm development at early stages and even result in embryo lethality. For example, *Emp5* encodes a mitochondrion-targeted DYW-type PPR protein, the *emp5* mutants exhibit abortion of embryo and endosperm development at early stages in maize [74]. Loss-of-function of the mitochondrial P-type PPR protein EMP10 severely disturbs embryo and endosperm development, resulting in empty pericarp or papery seeds in maize [78]. Additionally, the P-type protein PPR5 was recently identified as a regulator required for endosperm development in rice, the *ppr5* mutants develop small starch grains [106].

## 5. Mechanisms of PPR Protein-Mediated RNA Processing in Plant Organelles

Mostly, above-mentioned functions of PPR proteins in plant growth, development and stress responses are caused by the disturbed organellar gene expression pattern [107]. As a large family of RNA-binding proteins, some PPR proteins have shown to be required for post-transcriptional RNA processing events in plant mitochondria and chloroplasts [4]. These observations lead to an increasing interest towards understanding the nature and details of PPR protein-mediated RNA processing, and some progress has been made in the last decade [108], in particular, the mechanisms of PPR proteins involved in C-to-U RNA editing and in group II intron splicing.

### 5.1. Molecular Mechanisms of PPR Proteins in C-to-U RNA Editing

C-to-U RNA editing is highly prevalent in plant organelles, which converts cytidines to uridines at specific sites in transcripts of chloroplasts and plant mitochondria. The chemical nature of C-to-U RNA editing is through a site-specific cytidine deamination reaction [109]. To date, many E/E+ or DYW-subclass PPR proteins have been characterized as C-to-U RNA editing factors for one, several or multiple editing sites in plant organelles [84,85]. The DYW domain of some PPR proteins contains a conserved zinc-binding motif (HxE(x)_n_CxxC) in common with cytidine deaminase [110], and thus it has been proposed as the best candidate to catalyze deamination. Experimental support for this hypothesis came from mutagenesis experiments that modified the deaminase signature sequence. Mutation of several residues in cytidine deaminase signature of DYW1 significantly decreases the zinc-binding capacity and abolishes the editing of *ndhD*-1 [111]. The glutamate residue is required for the cytidine deaminase-catalyzed reactions. Mutagenesis of glutamate residue in the zinc-binding motif of OTP84 and CREF7 leads to loss of their editing activity at their cognate editing sites [112]. Recently, the DYW-PPR proteins called PPR56 and PPR65 from *Physcomitrella patens* were shown to drive C-to-U RNA editing individually at their corresponding targets in *E. coli* [113], and the purified recombinant PPR65 was found to exhibit the function of editing at its target site in synthetic RNAs [114], indicating a single DYW-deaminase domain-containing PPR protein is sufficient for editing its defined RNA targets within the bacterial cells and in vitro. In 2021, Takenaka et al. analyzed the structures of a DYW domain, they found that it contains a cytidine deaminase domain and a characteristic DYW motif, with catalytic and structural zinc atoms, respectively. Moreover, the DYW motif is stabilized by a zinc atom and functions in the control of zinc-mediated catalysis by altering the coordination setting around the catalytic zinc atom [115]. These results reinforce the hypothesis that the DYW domain functions as cytidine deaminase that carries out C-to-U RNA editing in plant organelles.

E and E+-type PPR proteins lack the DYW domain that contributes to the catalytic reaction of C-to-U RNA editing, and their exact function in the RNA-editing process remains unclear. Some studies rather suggest that they can form protein complexes with a DYW-type protein for a complete editing event. The first of these PPR editing factors to be identified is an E-type PPR protein CRR2. To edit chloroplast *ndhD*-2 site, CRR2 physically interacts with DYW1 that carries very few PPR motifs, no canonical E and E+ domains but end with a DYW domain [116]. Where, in which CRR2 acts as an editing site recognition factor, while DYW1 has key features of an editing deaminase [111]. DYW2 is a DYW-type PPR protein recently shown to be required for the editing of more than 100 sites in mitochondria and chloroplasts [117], and almost all of the DYW2 dependent editing sites are targets of E+-type PPR proteins, such as PGN, MEF37, OTP90 [118], etc. The functional association and physical interactions of DYW2 and E+-type PPR proteins suggest that DYW2 complements missing DYW domains in E+-type PPR proteins [117,118,119].

C-to-U RNA editing in plant organelles is performed by an editosome composed of RNA and protein factors. In addition to PPR proteins, several other protein factors were shown to be part of RNA editosome, and some of these protein factors were found to physically interact with each other or with PPR proteins. Previous studies indicate that the multiple organellar RNA editing factors (MORFs) can interact with the PLS-class PPR proteins and participate in RNA editing. MORF8 interacts with the DYW-type proteins RARE1 and MEF10 to participate in the C-to-U RNA editing in Arabidopsis chloroplast *accD* and mitochondrial *nad2* transcripts, respectively [120,121]. ZmMORF8 interacts with DYW-type proteins EMP5 and EMP21, and are together required for editing at multiple sites in maize mitochondria [85]. In addition, other MORF proteins have been shown to be involved in C-to-U RNA editing in plant mitochondria or chloroplasts, and physically interact with PLS-class PPR proteins [91,122]. RNA binding activity has been reported for PPR proteins, and a PPR-RNA recognition code has been elucidated and experimentally validated [123]. The PLS-class PPR proteins recognize and bind to RNA sequences upstream of the edited cytidine residues, whereas the MORF proteins mediate their RNA-binding activity [124]. For instance, MORF9 binding induced significant compressed conformational changes of PLS-class PPR protein, leading to the increase in the RNA-binding activity of PPR proteins [125].

These data indicate that PPR proteins serve as the primary recognition factors that single out specific cytidines to be converted into uridines in organellar transcripts of higher plants. These data also highlight that the DYW domain is the likely cytidine deaminase performing the cytidine deamination reaction (Figure 2A).

### 5.2. Molecular Mechanisms of PPR Proteins in Group II Intron Splicing

Group II introns commonly exist in organellar genomes of land plants, with more than 20 in the mitochondrial and chloroplast genomes of Arabidopsis, rice, and maize. Because of the loss of the ability to undergo self-splicing, the splicing of organellar group II introns in higher plants relies on various protein cofactors coming from different families and encoded by nuclear and organellar genomes [126,127]. So far, some P-class PPR proteins have been shown to be required for the splicing of plant mitochondrial and chloroplast introns [4], such as ABO5, MTL1, TANG2, OTP439, OTP43, and SLO3 in Arabidopsis [33,96,128,129,130], DEK2, DEK35, DEK37, EMP8, EMP10, EMP16, PPR14, EMP24, EMP25, and PPR-SMR1 in maize [52,54,56,76,78,82,89,93,131], as well as OsNPPR1, OsNPPR3, FLO10, and PPR5 in rice [106,132,133,134]. These PPR proteins are involved in the splicing of one, several or multiple introns in plant mitochondria or chloroplasts. For example, DEK37 [56] in maize, MID1 [104] in Arabidopsis, and FLO10 [133] in rice are specifically involved in the splicing of a single intron in mitochondria or chloroplasts. EMP8 [76] and PPR14 [89] participate in the splicing of several mitochondrial introns in maize, while PPR-SMR1 in maize is required for the splicing of nearly 75% of the mitochondrial introns [93]. Besides, some PLS-class PPR proteins that are usually characterized to function in RNA editing, are also implicated in RNA splicing. SLO4 was reported to affect *nad2* intron 1 splicing in Arabidopsis [135]. PpPPR43 influences the splicing efficiency of *cox1* intron 3 in *Physcomitrella patens* [136].

Although some PPR proteins acted as splicing cofactors in plant organelles have been identified, how these PPR proteins function in the splicing of plant organellar group II introns is not well understood. The recruitment of PPR proteins and other splicing factors is generally required for the splicing of a group II intron in plant organelles. In chloroplasts, the PPR proteins PPR4 and EMB2654 [101,137,138], the CRM (chloroplast RNA splicing and ribosome maturation) protein CAF2 [139], the PTH (peptidyl-tRNA hydrolase) protein CRS2 [140], and the DEAD-box RNA helicase protein RH3 [141] are required for the removal of intron 1 from *rps12* pre-mRNA, and they are all found in a large ribonucleoprotein complex. Similarly, in mitochondria, three PPR proteins, MISF68 [142], ZmSMK9 [68], and PPR-SMR1 [93], and a ribosomal protein uL18-L1 [143] participate in the splicing of *nad5* intron 1. In addition, the PPR protein PPR14 was reported to physically interact with a PPR protein PPR-SMR1 and a CRM-domain protein Zm-mCSF1, and they are required for the splicing of mitochondrial *nad2* intron 3 in maize [89,93]. As members of RNA-binding protein families, several P-class PPR proteins have been found to bind to their dependent splicing introns. For example, the chloroplast-localized THA8 is required for the splicing of *ycf3* intron 2 and *trnA* intron in maize [144], and its orthologues from *Brachypodium distachyon* can bind to *ycf3* intron 2 in vitro through multiple purine-rich sequences distributed in intron [145]. PPR4 and EMB2654 are known to be involved in the splicing of the chloroplast *rps12* intron 1 in maize and Arabidopsis, respectively [101,146]. RNA footprinting and EMSA experiments indicated that the Arabidopsis ortholog of PPR4 binds to a sequence near the 5′ end of *rps12* intron 1, while EMB2654 binds to a sequence near the 3′ end of *rps12* intron 1 [101,146]. By associating with their target intron, these PPR proteins and other splicing factors could form ribonucleoprotein complexes that help intron folding and stabilize their structure in an active form [146,147].

Taken together, these findings strongly suggest that PPR proteins participate in the splicing of group II intron in plant organelles might by binding to a specific RNA sequence in their target intron and interacting with other splicing factors to form a large complex that is required for the splicing performed (Figure 2B), but how these proteins collaborate or whether they act independently remains to be elucidated.

## 6. Conclusions and Perspectives

Although there is more evidence indicating that PPR proteins specifically take part in multiple aspects of RNA processing in plant organelles [4], only a few PPR proteins have been functionally characterized in contrast to the vast majority of PPR members. Moreover, computational and structural studies have provided new insights into RNA recognition and cytidine deamination reaction by PPR proteins in C-to-U RNA editing events [115,123,148]. However, little is known about the mechanistic details of PPR proteins involved in other RNA processing events. Therefore, there is a need to promote a more comprehensive understanding of PPR proteins. Firstly, why are PPR proteins particularly prevalent in land plants, and display diverse functions in organellar RNA processing? Secondly, how do PPR proteins interact with each other or other proteins to achieve their final functionality, and how do these interactions contribute to PPR activities in organellar RNA processing? Finally, what roles do PPR proteins play in the regulatory network of nuclear-cytoplasmic interaction in land plants? Maybe plenty of evolutionary analyses will provide deeper insight into their subtle classification and distribution. Functional identification of other PPR proteins and further analysis of co-expressed PPR genes will reveal whether they also play roles in organellar RNA processing or whether they are involved in other steps of RNA expression in plant organelles, and will allow us to define the regulatory network of nuclear-cytoplasmic interaction in land plants. Experimental identification of RNA targets of PPR proteins and crystal structure analyses of protein–RNA complexes are however necessary to elucidate the action mode of PPR proteins in detail. Besides, more attention should be paid to PPR interacting proteins that could directly affect the PPR protein effects, such as MORF proteins [47]. These studies will help to better clarify the mechanisms of PPR proteins, and also help to understand the details of plant organellar RNA processing.

## Figures and Tables

**Figure 1 ijms-22-11274-f001:**
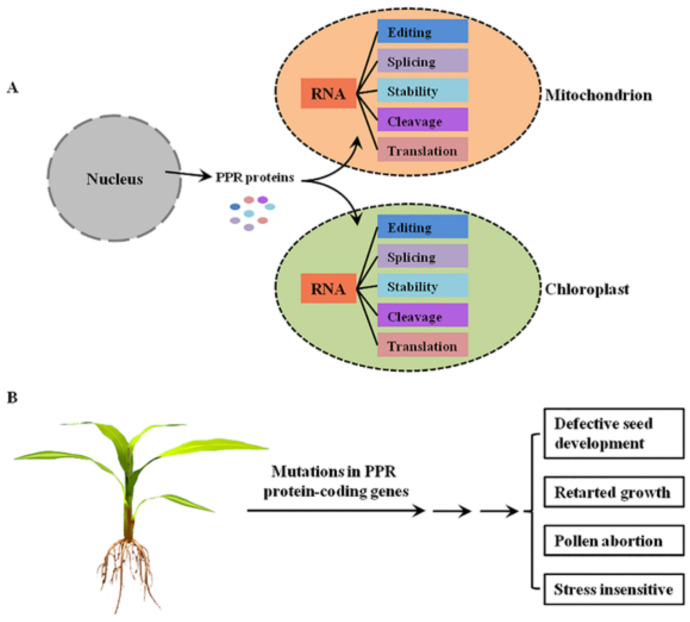
The functions of PPR proteins in plants. (**A**) The molecular functions of PPR proteins in plant mitochondria and chloroplasts. PPR proteins are encoded by nuclear genes, translated in the cytoplasm, and then imported into mitochondrion or chloroplast to mediate multiple steps of RNA processing. (**B**) The main growing and developmental phenotypes of plant mutants of PPR protein-coding genes.

**Figure 2 ijms-22-11274-f002:**
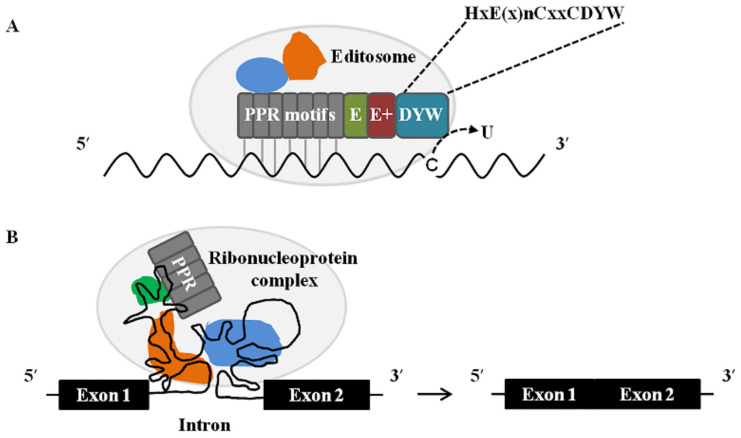
Proposed model for the molecular mechanisms of PPR proteins in plant organellar RNA editing and RNA splicing. (**A**) A model for the involvement of a DYW-type PPR protein in C-to-U RNA editing. C-to-U RNA editing in plant mitochondria and chloroplasts is carried out by an editosome composed of proteins and RNA. PPR motifs bind to its target RNA according to the PPR-RNA recognition code [123], which can be influenced by other proteins (shown in blue and orange). The DYW domain containing a cytidine deaminase signature (listed as HxE(x)nCxxC) converts the target cytidine (C) to uridine (U). (**B**) A model for the involvement of a P-class PPR protein in intron splicing. The splicing of introns in plant mitochondria and chloroplasts is proposed to be mediated by a ribonucleoprotein complex, where PPR proteins and other RNA-binding proteins (shown in green, blue, and orange) bind to their target intron to help its folding and stabilizing in an active form.

**Table 1 ijms-22-11274-t001:** Selected functionally characterized PPR proteins essential for seed development in maize and Arabidopsis.

Species	Subcellular Localization	Mutant Phenotype	Protein Name	PPR Class	Function(s)	References
Maize	Mitochondrion	*dek*	DEK2	P	RNA splicing, *nad1* intron 1	[52]
	Mitochondrion	*dek*	DEK10	PLS	RNA editing, *nad3*-61, 62, and *cox2*-550	[53]
	Mitochondrion	*dek*	DEK35	P	RNA splicing, *nad4* intron 1	[54]
	Mitochondrion	*dek*	DEK36	PLS	RNA editing, *atp4*-59, *nad7*-383, and *ccmFN*-302	[55]
	Mitochondrion	*dek*	DEK37	P	RNA splicing, *nad2* intron 1	[56]
	Mitochondrion	*dek*	DEK39	PLS	RNA editing, *nad3*-247 and *nad3*-275	[57]
	Mitochondrion	*dek*	DEK40	PLS	RNA editing, *cox3*-314, *nad2*-26, and *nad5*-1916	[58]
	Mitochondrion	*dek*	DEK41/ DEK43	P	RNA splicing, *nad4* intron 1 and 3	[59,60]
	Mitochondrion	*dek*	DEK46	PLS	RNA editing, D5-C22 of *nad7* intron 3 and 4	[61]
	Mitochondrion	*dek*	DEK53	PLS	RNA editing, multiples sites	[62]
	Mitochondrion	*dek*	DEK55	PLS	RNA splicing, *nad4* intron 1 and 3;RNA editing, multiple sites	[63]
	Mitochondrion	*dek*	DEK605	PLS	RNA editing, *nad1*-608	[64]
	Mitochondrion	*smk*	SMK1	PLS	RNA editing, *nad7*-836	[65]
	Mitochondrion	*smk*	SMK4	PLS	RNA editing, *cox1*-1489	[66]
	Mitochondrion	*smk*	SMK6	PLS	RNA editing, *nad1*-740, *nad4L*-110, *nad7*-739, and *mttB*-138, 139	[67]
	Mitochondrion	*smk*	ZmSMK9	P	RNA splicing, *nad5* intron 1 and 4	[68]
	Mitochondrion	*smk*	PPR2263	PLS	RNA editing, *nad5*-1550 and *cob*-908	[69]
	Mitochondrion	*smk*	MPPR6	P	Translation, *rps3* mRNA	[70]
	Mitochondrion	*dek/smk*	PPR20	P	RNA splicing, *nad2* intron 3	[71]
	Mitochondrion	*smk*	PPR78	P	RNA stabilization, *nad5* mature mRNA	[72]
	Mitochondrion	*emp*	EMP4	P	Expression of mitochondrial transcripts	[73]
	Mitochondrion	*emp*	EMP5	PLS	RNA editing, multiple sites	[74]
	Mitochondrion	*emp*	EMP7	PLS	RNA editing, *ccmFN*-1553	[75]
	Mitochondrion	*emp*	EMP8	P	RNA splicing, *nad1* intron 4, *nad2* intron 1, and *nad4* intron 1	[76]
	Mitochondrion	*emp*	EMP9	PLS	RNA editing, *ccmB*-43 and *rps4*-335	[77]
	Mitochondrion	*emp*	EMP10	P	RNA splicing, *nad2* intron 1	[78]
	Mitochondrion	*emp*	EMP11	P	RNA splicing, *nad1* intron 1, 2, 3, and 4	[79]
	Mitochondrion	*emp*	EMP12	P	RNA splicing, *nad2* intron 1, 2, and 4	[80]
	Chloroplast	*smk*	qKW9	PLS	RNA editing, *NdhB*-246	[81]
	Mitochondrion	*emp*	EMP16	P	RNA splicing, *nad2* intron 4	[82]
	Mitochondrion	*emp*	EMP17	PLS	RNA editing, *c**cmF_C_*-799 and *nad2*-677	[83]
	Mitochondrion	*emp*	EMP18	PLS	RNA editing, *atp6*-635 and *cox2*-449	[84]
	Mitochondrion	*emp*	EMP21	PLS	RNA editing, multiple sites	[85]
	Mitochondrion	*emp*	EMP32	P	RNA splicing, *nad7* intron 2	[86]
	Mitochondrion	*emp*	EMP602	P	RNA splicing, *nad4* intron 1 and 3	[87]
	Mitochondrion	*emp*	EMP603	P	RNA splicing, *nad1* intron 2	[88]
	Mitochondrion	*emp*	PPR14	P	RNA splicing, *nad2* intron 3, *nad7* intron 1 and 2	[89]
	Mitochondrion	*emp*	PPR18	P	RNA splicing, *nad4* intron 1	[90]
	Mitochondrion	*emp*	PPR27	PLS	RNA editing, multiple sites	[91]
	Mitochondrion	*emp*	PPR101	P	RNA splicing, *nad5* intron 1 and 2	[92]
	Mitochondrion	*emp*	PPR231	P	RNA splicing, *nad5* intron 1, 2, 3 and *nad2* intron 3	[92]
	Mitochondrion	*emp*	PPR-SMR1	P	RNA splicing, multiple introns	[93]
	Chloroplast	*emb*	PPR8522	P	RNA transcription, nearly all chloroplast-encoded genes	[94]
	Chloroplast	*emb*	EMB-7L	P	RNA splicing, multiple introns	[95]
Arabidopsis	Mitochondrion	*dek*	OTP43	P	RNA splicing, *nad1* intron 1	[96]
	Mitochondrion	*smk*	PPR19	P	RNA stabilization, *nad1* mature mRNA	[97]
	Mitochondrion	*emp*	BLX	PLS	RNA editing, multiple sites; RNA splicing, *nad1* intron 4 and *nad2* intron 1	[98]
	Chloroplast	*emb*	AtPPR2	P	RNA translation	[99]
	Chloroplast	*emb*	ECD2	P	RNA splicing, *ndhA*, *ycf3* intron 1, *rps12* intron 2 and *clpp* intron 2	[100]
	Chloroplast	*emb*	EMB2654	P	RNA splicing, *rps12* intron 1	[101]
	Mitochondrion	*emb*	EMB2794	P	RNA splicing, *nad2* intron 2	[102]
	Nucleus	*emb*	GRP23	P	RNA transcription	[103]
	Mitochondrion	*emb*	MID1	P	RNA splicing, *nad2* intron 1	[104]
	Chloroplast	*emb*	PMD3	P	RNA splicing, *trnA*, *ndhB*, and *clpP-1*	[105]

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
