# Peer review of "Functions of PPR Proteins in Plant Growth and Development"

_ijms, 2021, doi:10.3390/ijms222011274_

Round 1

Reviewer 1 Report

The manuscript cover many important artices  in this field. 

Author Response

Dear Reviewer,

We appreciate very much for your comments on our manuscript (ijms-1403755). We have studied these comments carefully and they are all valuable for revising and improving our manuscript. Responds to these comments are as following.

English language and style are fine/minor spell check required.

Response:

Thanks for the suggestion. We have paid high attention on your suggestion and modified the language with the help of our native English-speaking colleague. In the revised version, changes to our manuscript have been marked up using the “Track Changes” function.

Reviewer 2 Report

The manuscript "Functions of PPR Proteins in Plant Growth and Development." has been submitted as a review by Li et al.

The manuscript is well written and informative. I have only a few points that have to be addressed.

1.) Please consider to add more general aspects concerning the term "PPR motif". In addition to the used "pentatricopeptide-repeat", you can find also the PPR standing for "proline-proline-arginine"-domain or as a variant of the PXXP-motif in the literature. Please briefly specify the definition and exclude other variants.

2.) The manuscript presents PPR domains as RNA binding domains. Please specify which kind of RNA is meant here (both in the text and figures or figure legends).

3.) Even though there is only one table, please name it Table 1.

4.) It supports the presentation, if the table would not be distributed between two pages. If there is no other possibility to do so, maybe you could consider to start with the shorter A. thaliana-part and then present the entire maize part on the second page.

5.) Please add a bit more descriptive text to the figure legends.

6.) I understand that this is a plant-centered review. However, it would be interesting to add at least a few cross-references to related PPR proteins in animals, if relevant enough.

Author Response

Dear Reviewer,

We appreciate very much for your comments on our manuscript (ijms-1403755). We have studied these comments carefully and they are all valuable for revising and improving our manuscript. Responds to these comments are as following.

1.) Please consider to add more general aspects concerning the term “PPR motif”. In addition to the used “pentatricopeptide-repeat”, you can find also the PPR standing for “proline-proline-arginine”-domain or as a variant of the PXXP-motif in literature. Please briefly specify the definition and exclude other variants.

Response:

Thanks for your suggestion. We agree that the “proline-proline-arginine”-domain, one kind of poly-proline-arginine (PXXXPR) motif, is also abbreviated to “PPR motif”. The definition of PPR proteins has been modified as “Pentatricopeptide repeat (PPR) proteins are characterized by the presence of tandem arrays of a degenerate 35-amino-acid repeat motif, PPR motif [1]. (Lines 21-22).” In which, we point out that PPR stands for “pentatricopeptide repeat” that means 35-peptide repeat and is consistent with “PPR motif (35-amino-acid repeat motif)”. We consider that “PPR motif” in our revised manuscript is no longer confused with the “PPR motif” standing for “proline-proline-arginine”-domain.

2.) The manuscript presents PPR domains as RNA binding domain. Please specify which kind of RNA is meant here (both in the text and figures or figure legends).

Response:

Thanks for the suggestion. RNA binding activity has been reported for PPR proteins, and a PPR-RNA recognition code has been elucidated and experimentally validated. Some PPR proteins have shown to bind to RNA sequences of precursor mRNA or mature mRNA. Several PPR proteins have been identified to be involved in the processing of tRNA or rRNA, and they may also function by binding to their target tRNA or rRNA. Therefore, we feel sorry that we should not specify the kind of RNA bound by PPR proteins. Thanks for your suggestion again.

3.) Even though there is only one table, please name it Table 1.

Response:

According to your suggestion, the table has been renamed to Table 1. (Lines 155-156)

4.) It supports the presentation, if the table would not be distributed between two pages. If there is no other possibility to do so, maybe you could consider to start with the shorter A. thaliana-part and then present the entire maize part on the second page.

Response:

It is a good suggestion. If the table starts from the Arabidopsis part, the entire maize part still could not be completely presented on the next page, while it will cover three pages. We are looking forward to communicate with the editor to present the maize part on one page and the Arabidopsis part on the next page.

5.) Please add a bit more descriptive text to figure legends.

Response:

It is a helpful suggestion. We have added more descriptive text to figure legends. The figure legend in Figure 1 has been modified as “Figure 1. The functions of PPR proteins in plants. (A) The molecular functions of PPR proteins in plant mitochondria and chloroplasts. PPR proteins are encoded by nuclear genes, translated in the cytoplasm, and then imported into mitochondrion or chloroplast to mediate multiple steps of RNA processing. (Lines 39-42)”

The figure legend in Figure 2 has been modified as “Figure 2. Proposed model for the molecular mechanisms of PPR proteins in plant organellar RNA editing and RNA splicing. (A) A model for the association of a DYW-type PPR protein in C-to-U RNA editing. C-to-U RNA editing in plant mitochondria and chloroplasts is carried out by an editosome composed of proteins and RNA. PPR motifs bind to its target RNA according to the PPR-RNA recognition code [123], which can be influenced by other proteins (shown in blue and orange). The DYW domain containing a cytidine deaminase signature (listed as HxE(x)nCxxC) converts the target cytidine (C) to uridine (U). (B) A model for the association of a P-class PPR protein in intron splicing. The splicing of introns in plant mitochondria and chloroplasts is proposed to be mediated by a splicesome, where PPR proteins and other RNA-binding proteins (shown in green, blue and orange) bind to their target intron to help its folding and stabilizing in an active form. (Lines 253-262)”

6.) I understand that this is a plant-centered review. However, it would be interesting to add at least a few cross-references to related PPR proteins in animals, if relevant enough.

Response:

Thanks for the suggestion. PPR proteins form a large family in higher plants, with more than 400 members in most species. More and more studies have indicated that PPR proteins are mostly targeted to plant mitochondria or chloroplasts, where they are involved in multiple aspects of RNA processing. In comparison, there are only a few PPR proteins in animals, such as 2 in Drosophila and 6 in human, as well as little is known about their functions. Considering our manuscript mainly reviews functions of PPR proteins in plant growth and development, which has no enough relevance to PPR proteins in animals. It is not appropriate to list references related to animal PPR proteins in our manuscript. Thanks for your suggestion again.

Moderate English changes required.

Response:

Thanks for the suggestion. We have paid high attention on your suggestion and modified the language with the help of our native English-speaking colleague. In the revised version, changes to our manuscript have been marked up using the “Track Changes” function.
